# Resistance to TGFβ suppression and improved anti-tumor responses in CD8+ T cells lacking PTPN22

Rebecca J. Brownlie [1,2], Celine Garcia[1], Mate Ravasz[1], Dietmar Zehn[3], Robert J. Salmond [1,2] & Rose Zamoyska [1]

Transforming growth factor β (TGFβ) is important in maintaining self-tolerance and inhibits T cell reactivity. We show that CD8+ T cells that lack the tyrosine phosphatase *Ptpn22*, a major predisposing gene for autoimmune disease, are resistant to the suppressive effects of TGFβ. Resistance to TGFβ suppression, while disadvantageous in autoimmunity, helps *Ptpn22*−/− T cells to be intrinsically superior at clearing established tumors that secrete TGFβ. Mechanistically, loss of Ptpn22 increases the capacity of T cells to produce IL-2, which overcomes TGFβ-mediated suppression. These data suggest that a viable strategy to improve anti-tumor adoptive cell therapy may be to engineer tumor-restricted T cells with mutations identified as risk factors for autoimmunity.

[1] Institute of Immunology and Infection Research, University of Edinburgh, Ashworth Laboratories, Charlotte Auerbach Road, Edinburgh EH9 3FL, UK. [2] Leeds Institute of Cancer and Pathology, University of Leeds, Wellcome Trust Brenner Building, St James's University Hospital, Leeds LS9 7TF, UK. [3] School of Life Sciences Weihenstephan, Technical University Munich, Weihenstephaner Berg 3, Freising 85354, Germany. Correspondence and requests for materials should be addressed to R.J.S. (email: r.j.salmond@leeds.ac.uk) or to R.Z. (email: rose.zamoyska@ed.ac.uk)

T cell immune tolerance is maintained in the periphery by a combination of extrinsic factors, including the action of regulatory T cells (Treg) and suppressive cytokines such as transforming growth factor β (TGFβ). TGFβ cytokines are positive and negative regulators of proliferation and differentiation in many cell lineages, and play a key role in maintaining peripheral T cell tolerance[1]. TGFβ signaling was shown to be essential to maintain peripheral T cell quiescence in vivo[2–5] and to be a negative regulator of T cell proliferation in vitro[6]. Mice whose T cells are unresponsive to TGFβ signals either by expressing a dominant negative TGFβRII transgene[2,4], or through specific T cell deletion of the TGFβ receptor, Tgfbr1[3,5] die from fatal lymphoproliferative disease. In the absence of TGFβ signaling, peripheral CD4+ and CD8+ T cells become activated in a cell-intrinsic fashion, most likely due to homeostatic expansion from the lymphopenic environment shortly after birth, and develop effector function and pathogenicity[7,8]. Thus, TGFβ signals are particularly important for the regulation of the TCR-mediated, weak, self-ligand interactions that maintain peripheral T cell homeostasis.

TGFβ signaling is important also in restricting T cell responses to tumors[9,10]. In the context of CD8+ T cells, TGFβ targets the transcription of key effector molecules and interferes with their ability to kill tumor cells[10]. These observations have led to the suggestion that targeting TGFβ sensitivity, for example, by inactivating the TGFβ receptor in T cells, could be beneficial in anti-tumor T cell therapy[11].

There are several parallels between the induction of T cell autoimmunity and effective tumor immunosurveillance[12,13]. T cells in autoimmune disease display functional characteristics that, in a tumor setting, may be beneficial. For example, auto-reactive T cells respond effectively to poorly immunogenic antigens and are resistant to immune-regulatory mechanisms[2,8]. Similarly, in order to mount effective anti-cancer responses, T cells need to overcome a strongly inhibitory tumor micro-environment, characterized by the presence of Tregs and high levels of inhibitory cytokines such as TGFβ. Indeed, TGFβ is expressed by a wide range of cancer cell types (https://portals.broadinstitute.org/ccle/home) as well as by tumor-infiltrating immune cells such as myeloid-derived suppressor cells[14].

Genome-wide association studies have shown that the hematopoietic tyrosine phosphatase, PTPN22, has a commonly expressed variant, PTPN22R620W, that is highly associated with predisposition to a number of autoimmune diseases[15,16]. PTPN22 regulates Ag-specific T- and B-cell responses and some pattern recognition signaling pathways in myeloid cells that lead to production of type 1 interferons[15]. We showed previously that mouse Ptpn22−/− T cells are partially resistant to wild-type Treg-mediated suppression[17] and are more responsive to low-affinity antigens[18]. Given the critical dual role of TGFβ in limiting autoimmune and anti-tumor responses, we investigated the impact of PTPN22 deficiency on the responses of CD8+ T cells to this key inhibitory cytokine. Furthermore, we sought to test the hypothesis that PTPN22 acts as a brake to limit the effectiveness of anti-tumor T cell responses. We show that Ptpn22−/− CD8 T cells are considerably more resistant to the suppressive effects of TGFβ than WT T cells. Concentrations of TGFβ that suppress proliferation and differentiation of effector cytokines in WT T cells show little inhibition of these processes in Ptpn22−/− CD8 T cells. Upon stimulation with both weak and strong agonists peptides Ptpn22−/− CD8 T cells produce more IL-2 than WT T cells and IL-2 interferes with the suppressive effect of TGFβ. Consequently upon adoptive transfer, Ptpn22−/− CD8 T cells are better able than WT CD8 T cells to control the growth of established tumors that secrete TGFβ. Importantly, the superior anti-tumor capacity of Ptpn22−/− CD8+ T cells is observed in response to both strong and weak antigens, the latter being a common trait of tumor-associated antigens, which can otherwise limit robust anti-tumor immune responses. These results suggest that targeting genes associated with susceptibility to autoimmunity may be a viable strategy to improve the efficacy of adoptive T cell immunotherapy.

## Results

**Ptpn22−/− T cells resist TGFβ-mediated suppression.** TGF-β was shown previously to suppress low-affinity T cell responses more effectively than high-affinity responses[8] and we showed a similar role for PTPN22 in limiting CD8+ T cell responses[18]. Using the OVA-specific TCR transgenic mouse, OT-1 on a Rag-1−/− background (hereafter called OT-1 T cells), for which a number of peptides have been characterized, which span a range of affinities[19], we examined the sensitivity of control and Ptpn22−/− OT-1 T cells to inhibition by TGFβ in vitro.

As reported previously[8], TGFβ markedly inhibited antigen-induced proliferation of OT-1 TCR transgenic T cells in a dose-dependent manner (Fig. 1a–c). This suppression occurred following stimulation both with strong agonist, SIINFEKL (N4) peptide and with weak agonist, SIITFEKL (T4) peptide. Enhanced proliferation of Ptpn22−/− OT-1 T cells compared to control OT-1 cells was particularly apparent in response to weak agonist peptide, T4, in the absence of TGFβ (Fig. 1a–c), as we reported previously[18]. Significantly, both N4- and T4-induced proliferation of Ptpn22−/− OT-1 T cells were extremely refractory to TGFβ inhibition (Fig. 1a–c) compared to control OT-1 T cells. In response to N4 peptide, no inhibition of Ptpn22−/− OT-1 T cell proliferation was seen at doses of up to 5 ng/ml TGFβ, while in response to T4 peptide, Ptpn22−/− OT-1 T cells required ~10-fold higher concentrations of TGFβ than control OT-1 T cells to show equivalent suppression of proliferation.

The mechanism of TGFβ-mediated suppression of T cell responses is not well understood. PTPN22 is known to target signaling molecules immediately downstream of the TCR, such as Lck and ZAP70, so it was unclear why Ptpn22−/− T cells would be more resistant than control T cells to TGFβ-mediated suppression. We asked at what point TGFβ suppression acted to inhibit control OT-1 T cells. Time-course analysis demonstrated that TGFβ inhibition of OT-1 T cell proliferation was apparent at day 3 (Fig. 1d), but in order to achieve this, TGFβ needed to be present from the start of culture. Delaying addition of TGFβ by 2 days failed to limit antigen-induced proliferation (Fig. 1e). Similarly, TGFβ-mediated inhibition of OT-1 T cell proliferation could be alleviated by an inhibitor to the TGFβ receptor, but only when added from the beginning of culture (Fig. 1f).

These data show that TGFβ acted early in T cell activation, as by 48 h after stimulation cells were resistant to the anti-proliferative effects of TGFβ. Accordingly, we analyzed activation marker and transcription factor expression at 24 h following stimulation of OT-1 T cells. In control OT-1 T cells, TGFβ limited the extent of weak agonist T4-induced upregulation of the IL-2 receptor-α chain (CD25), and key transcription factors Myc and IRF4, but less so Tbet, whereas Ptpn22−/− OT-1 T cells were refractory to these inhibitory effects of TGFβ (Fig. 2a, b). These data indicate that loss of PTPN22 counteracted TGFβ inhibition during the early phase of TCR engagement.

**Ptpn22−/− CD8+ T cells are superior at tumor clearance.** The resistance of Ptpn22−/− T cells to TGFβ-mediated suppression in vitro led us to ask whether these CD8+ T cells would be more effective for adoptive cell therapy against TGFβ-secreting tumors in vivo. First, we evaluated the ability of control and Ptpn22−/− OT-1 T cells to reject EL4 lymphoma cells (EL4-OVA) expressing

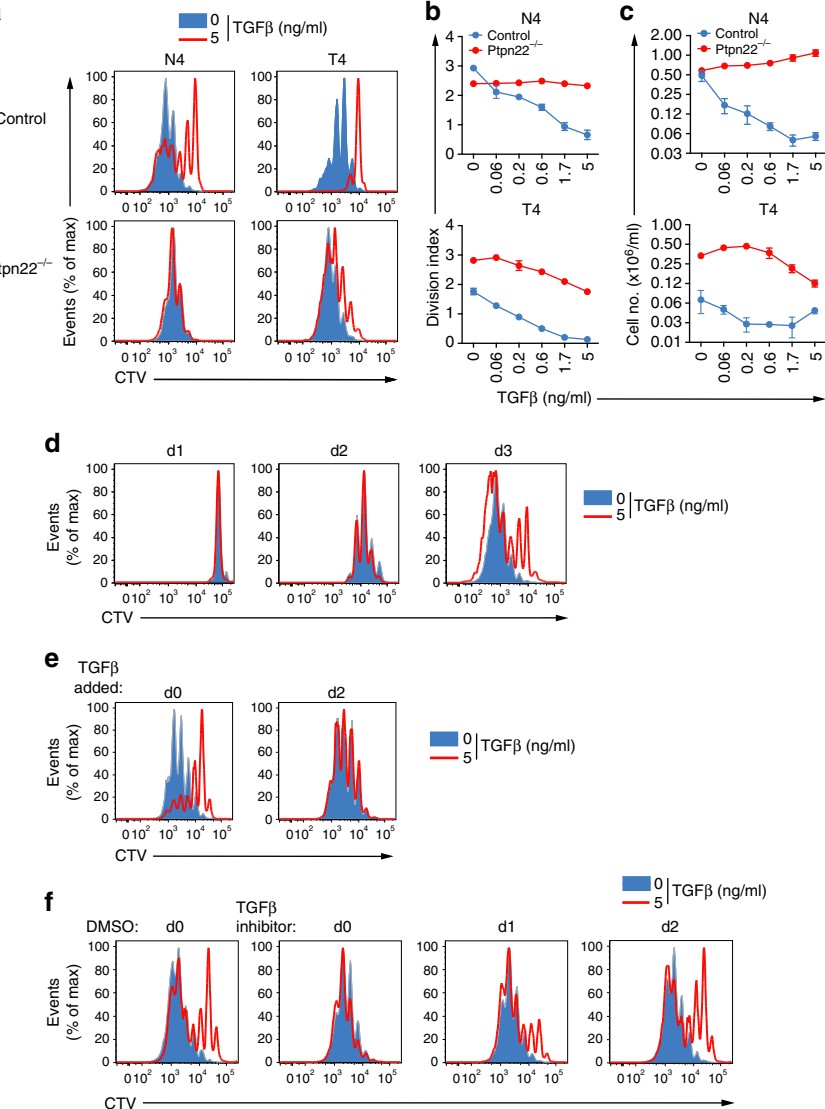

**Fig. 1** *Ptpn22* deficiency protects cells from TGFβ-mediated suppression. **a** Dilution of Cell Trace Violet (CTV) in control and *Ptpn22*−/− OT-1 cells was used to assess proliferation of cells stimulated with N4 or T4 peptide ($10^{-6}$M) for 3 days in the presence of varying doses of TGFβ (0–5 ng/ml). **b** Proliferation was quantified by calculation of division index (FlowJo). **c** Numbers of live control and *Ptpn22*−/− OT-1 cells recovered after stimulation was calculated using MACSquant software (mean ± s.d. for triplicate replicates). **d** TGFβ inhibition becomes apparent only after 2d of culture. Dilution of CTV from control OT-1 cells stimulated with T4 ($10^{-6}$ M) ± TGFβ (5 ng/ml) after d1, d2, or d3 of culture. **e** Dilution of CTV violet from control OT-1 cells stimulated with T4 ($10^{-6}$ M) ± TGFβ (5 ng/ml) added at d0 or d2 of culture. **f** Inhibition of TGFβR1 signaling, by addition of ALK-5/TGFβR1 inhibitor SB431542, at d0 interferes with TGFβ-mediated suppression of proliferation, but SB431542 is less effective when added on d1 or d2 of culture. Control OT-1 cells stimulated with T4 ($10^{-6}$ M) ± TGFβ (5 ng/ml) were cultured for 3d ± 5 μM SB431542 (or DMSO vehicle control) added on d0, d1, or d2 of culture. Representative histograms of CTV dilution analysed by flow cytometry are shown in **a**, **d**, **e**, **f**. Data are representative of 5 (**a**–**c**), 2 (**d**, **e**), and 3 (**f**) separate experiments

high-affinity N4 peptide as a tumor-specific antigen. EL4-OVA tumors were established subcutaneously for 5d before transfer of small numbers ($5 \times 10^4$/mouse) of either control or *Ptpn22*−/− naive OT-1 T cells. Tumor size was monitored for a further 7d after which mice were killed to prevent control animals exceeding permitted tumor volumes, and wet tumor mass measured. Strikingly, *Ptpn22*−/− OT-1 T cells were significantly superior to control cells in controlling growth of subcutaneous EL4-OVA tumors (*p < 0.05 using two-way ANOVA with Tukey's post test) (Fig. 3a). In particular, very few animals receiving *Ptpn22*−/− OT-1 T cells showed evidence of tumor mass over the time course following transfer, whereas a number of those receiving control OT-1 T cells showed ongoing tumor growth followed by regression toward the end of the experiment.

TGFβ is secreted by EL4 cells, so we asked to what extent TGFβ was preventing tumor rejection by OT-1 T cells[2,20–22]. Subcutaneous tumors were established as before, but with an EL4-OVA cell line engineered to express a soluble form of the TGFβRII (EL4-OVA STβRII)[10]. Adoptively transferred control OT-1 T cells were better able to reduce tumor growth when TGFβ was blocked, as shown previously[10], while *Ptpn22*−/− OT-1 T cells effected complete rejection of EL4-OVA STβRII cells (20/20 mice) (Fig. 3b). These data illustrate that while, not completely immune to TGFβ inhibition, *Ptpn22*−/− T cells are inherently superior in managing growth of TGFβ-secreting tumors.

Key to control of tumor mass is the ability of CD8+ T cells to kill tumor targets. TGFβ acts to inhibit the expression of several cytolytic gene products, thus hindering CD8 T cell cytotoxicity[10].

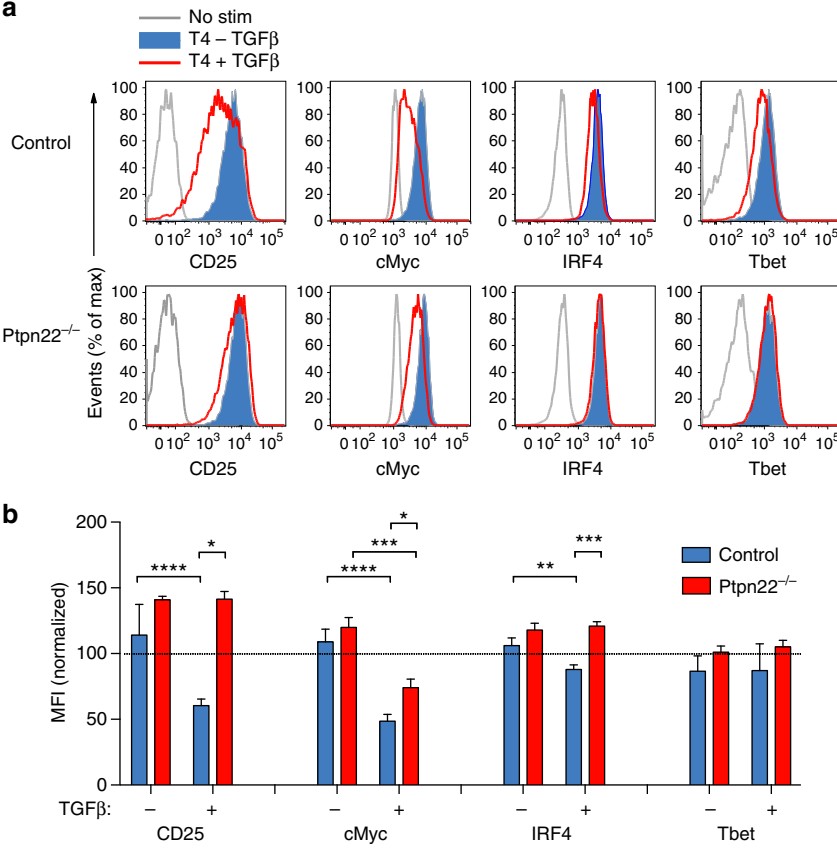

**Fig. 2** PTPN22 mediates the extent of TGFβ-mediated suppression of early activation markers. **a** Control and *Ptpn22*$^{-/-}$ OT-1 cells were cultured for 24 h with T4 peptide ($10^{-6}$ M) ± TGFβ (5 ng/ml). Activation was assessed by upregulation of CD25 and transcription factors c-Myc, IRF4, Tbet by flow cytometry. Representative histograms are shown. **b** Quantification of MFIs of peptide stimulated cells ± TGFβ 5 ng/ml as shown in **a** (mean and s.d. of three replicates and data representative of 5 experiments). *$p < 0.05$, **$p < 0.01$, ***$p < 0.001$, and ****$p < 0.0001$, as determined using two-way ANOVA with Tukey's post test for multiple comparisons

To better mimic responses to weak tumor antigens, we used ID8 murine ovarian carcinoma cells[23] that as a tumor-specific antigen expressed the weak agonist peptide, T4, which is ~70-fold less potent in stimulating OT-1 T cells than N4 peptide[24], and is of equivalent affinity to self-peptides that select the thymic repertoire[19]. We cultured ID8-T4 cells with control and *Ptpn22*$^{-/-}$ OT-1 T cells in vitro and monitored production of granzyme B (GzB) and IFNγ. Remarkably, while GzB was detectable in less than 10% of control cells, ~70% of *Ptpn22*$^{-/-}$ T cells were positive for GzB and a substantial proportion of the latter also expressed IFNγ in response to ID8-T4 cells alone. Blockade of TGFβ signaling by addition of a TGFβ-receptor inhibitor relieved the suppression of control T cells so that now ~40% expressed granzyme B while >90% of *Ptpn22*$^{-/-}$ cells became GzB$^+$ (Fig. 3c).

We then compared the ability of control and *Ptpn22*$^{-/-}$ OT-1 CD8$^+$ TCR transgenic T cells to respond to tumors expressing low-affinity TSA in vivo. ID8-T4 tumors were established in the peritoneum of recipient B6 (CD45.1/CD45.2F$_1$) mice for 28d before transfer of an equal mixture of control CD45.1$^+$ and *Ptpn22*$^{-/-}$ CD45.2$^+$ OT-1 T cells. Three days later, we found significantly enhanced proliferation of *Ptpn22*$^{-/-}$ OT-1 T cells compared to control cells in individual mice (Fig. 3d, **$p < 0.01$ using two-way ANOVA with Tukey's post test) showing that *Ptpn22*$^{-/-}$ T cells indeed proliferate more in response to the weak T4 antigen expressed by the tumor.

In order to non-invasively assess ID8 tumor growth in vivo, we transduced ID8-T4 cells with lentivirus-expressing Firefly

luciferase (Fluc), enabling us to monitor bioluminescence as a correlate of tumor mass. Tumors were established in the peritoneum of B6 mice prior to adoptive transfer of control or *Ptpn22*$^{-/-}$ OT-1 T cells, that were first activated in vitro. Tumor bioluminescence was analyzed 14d post T cell transfer. While control OT-1 T cells reduced tumor mass in a number of recipient mice, this effect did not reach statistical significance (Fig. 3e). By contrast, adoptive cell transfer of *Ptpn22*$^{-/-}$ OT-1 cells resulted in a significant decrease in luminescence signal, and in the majority of mice, a complete rejection of tumor load (Fig. 3e, *$p < 0.05$ using two-way ANOVA with Tukey's post test).

Taken together, our data using EL4 lymphoma and ID8 ovarian carcinoma models indicate that *Ptpn22*$^{-/-}$ T cells generate a more effective anti-tumor responses to both high- and low-affinity TSA. This effect most likely stems from a combination of the enhanced TCR responsiveness and proliferation of cells lacking PTPN22 to both high- and low-affinity TSA and their reduced susceptibility to inhibitory cytokines such as TGFβ, permitting differentiation and expression of effector cytokines and the cytolytic machinery.

**Canonical TGFβR signaling is unaffected by loss of PTPN22.** We next assessed the mechanisms underlying the ability of *Ptpn22*$^{-/-}$ T cells to overcome TGFβ-mediated suppression. Surface TGFβRII expression (Fig. 4a) and canonical TGFβRII signaling, judged by TGFβ-induced phospho-SMAD (Fig. 4b), were identical in control and *Ptpn22*$^{-/-}$ OT-1 T cells.

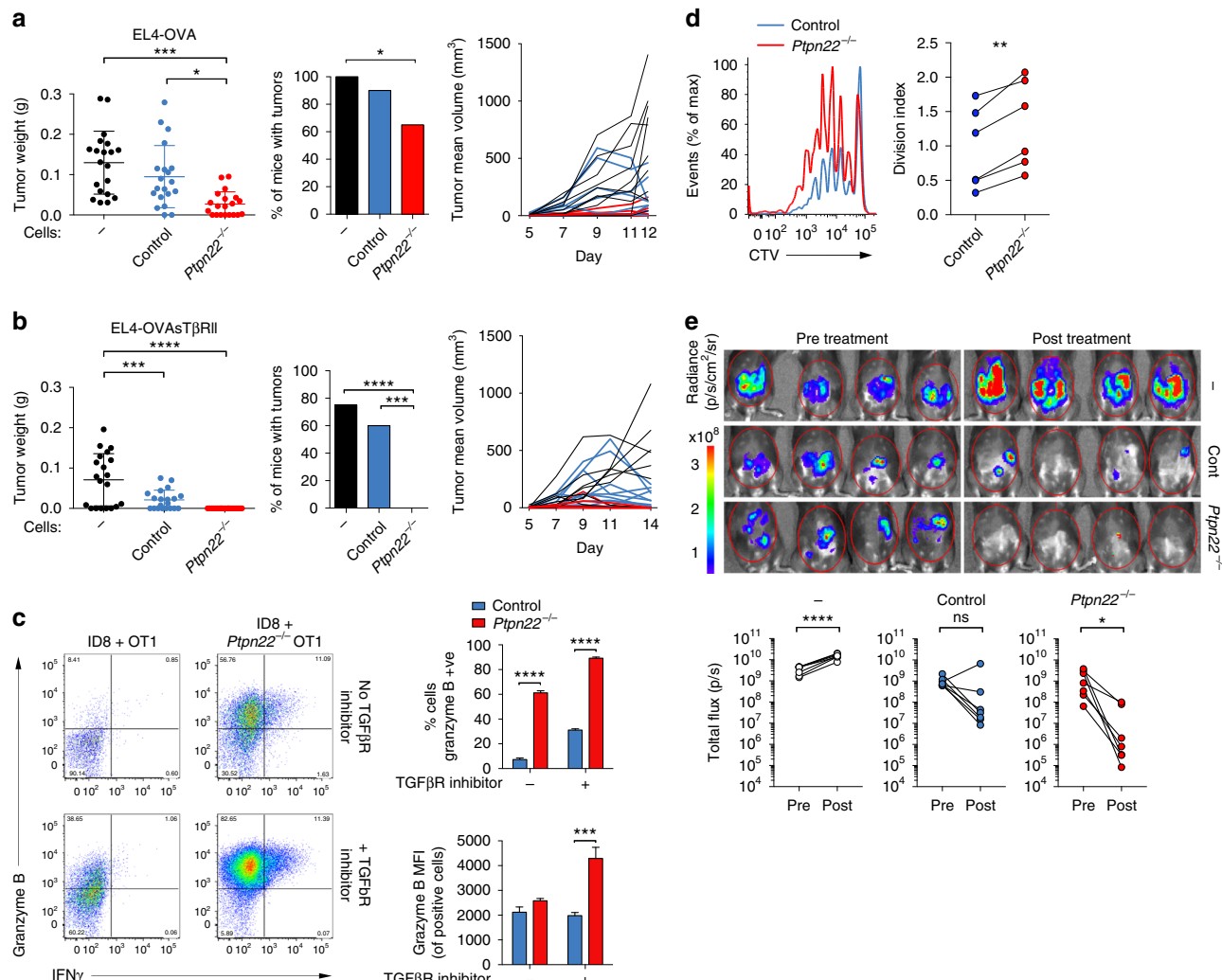

**Fig. 3** Loss of *Ptpn22* improves the response of CD8 T cells to tumors. **a**, **b** B6 mice were injected s.c. with $1 \times 10^6$ EL4-OVA (**a**) or EL4-OVAsTβRII cells (**b**). Five days later, $5 \times 10^4$ control or *Ptpn22*$^{-/-}$ OT-1 cells were injected i.v. Tumor weight (left panel) and tumor incidence (middle panel) is shown (d12 for EL4-OVA, d14 for EL4-OVAsTβRII from two pooled experiments, *n* = 20/group). Right panel shows mean tumor volume of individual mice, which received no exogenous T cells (black lines), control OT-1 T cells (blue lines), or *Ptpn22*$^{-/-}$ OT-1 T cells (red lines) until termination of the experiment on d12 (**a**) or d14 (**b**) from one experiment (*n* = 10/group). **c** Control and *Ptpn22*$^{-/-}$ OT-1 cells were co-cultured with ID8 tumor cells ± ALK-5/TGFR1 inhibitor SB431542. After 2 days, Brefeldin A was added to cultures for 6 h and cells were stained for intracellular granzyme B and IFNγ analysis by flow cytometry. Representative dot plots and quantification (mean and s.d. of three replicates) are shown. **a–c** \**p* < 0.05, \*\**p* < 0.01, \*\*\**p* < 0.001, \*\*\*\**p* < 0.0001 as determined using two-way ANOVA with Tukey's post test for multiple comparisons and for incidence data \*\*\**p* < 0.001 and \*\*\*\**p* < 0.0001 as determined by χ²-test with Fisher's exact post test adjusting the *p* value for number of comparisons. Data are representative of three separate experiments. **d** Groups (*n* = 6) of CD45.1/2 B6 mice were injected i.p. with $1 \times 10^7$ ID8 cells, followed 28 days later with an i.p. injection of a mix of CTV-labeled control CD45.1$^+$ OT-1 and *Ptpn22*$^{-/-}$ CD45.2$^+$ OT-1 cells ($1 \times 10^6$ of each). Peritoneal lavage was harvested 3 days later and assessed for OT-1 T cell proliferation by flow cytometry. Division index was calculated using Flowjo software. \*\**P* < 0.01 as determined by Student's paired *t* test. Data are representative of two experiments. **e** Groups (*n* = 7) of B6 mice were injected i.p. with $5 \times 10^6$ ID8-T4-fluc2 cells and assessed for tumor establishment on d11 (pre treatment) by bioluminescence imaging. On d12, groups of mice received either no cells, $10 \times 10^6$ control OT-1 CTLs, or Ptpn22$^{-/-}$ OT-1 CTLs i.p. All mice were assessed for tumor growth by bioluminescence imaging on d26 (14 d post treatment). Example images of *n* = 4 mice/group are shown d11 and d26 (pre/post treatment). Graphs show bioluminescence signal intensity of all mice on d11 and d26 (pre/post treatment). \**p* < 0.05, \*\*\*\**p* < 0.0001, as determined by Student's paired *t* test. Data are representative of two experiments

Furthermore, in both genotypes, there was SMAD-dependent inhibition of granzyme B expression[20] and upregulation of CD44 expression by high-dose TGFβ (Fig. 4c). Recent analyses have shown that expression of the transcription factor FoxP1 is essential for TGFβ-mediated suppression of anti-tumor T cells[25], however, levels of FoxP1 expression by control and *Ptpn22*$^{-/-}$ T cells were comparable under basal and activated conditions (Fig. 4d).

In order to determine whether TGFβ directly affects PTPN22 function, we analyzed the phosphorylation of well-defined PTPN22 substrates ZAP70 and TCRζ[26] following TCR stimulation of control OT-1 T cells in the presence or absence of TGFβ. Importantly, TGFβ did not impair basal or TCR-stimulated levels of Zap70 Y493, TCRζ Y83, or ERK MAPKinase phosphorylation (Fig. 4e). Previous data have shown that PTPN22 associates with Csk, which may be important for its function in T cells[27]. However, co-immunoprecipitation of PTPN22 with Csk was unaffected by culture of OT-1 T cells with TGFβ (Fig. 4f). Together, these data show that PTPN22 does not impact directly upon canonical SMAD-dependent TGFβ receptor signaling

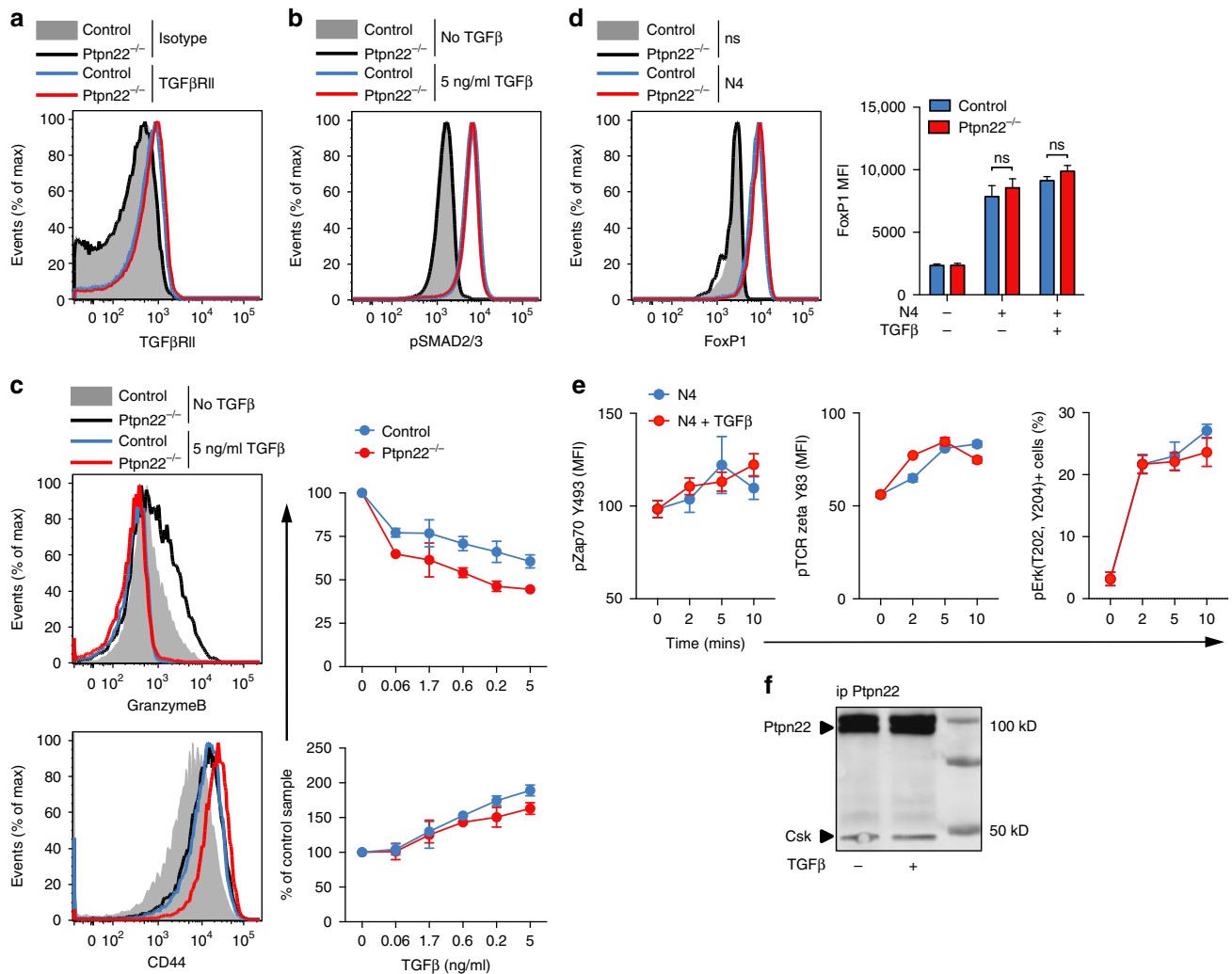

**Fig. 4** TGFβ signaling is not affected by the absence of PTPN22. Histogram overlays show equivalent expression in control and *Ptpn22*<sup>−/−</sup> OT-1 cells of **a** cell surface TGFβRII and **b** intracellular SMAD2/3 phosphorylation (pSMAD) after stimulation with 5 ng/ml TGFβ for 30 min. **c** The TGFβ-regulated genes granzyme B and CD44 were down- and upregulated, respectively, in both control and *Ptpn22*<sup>−/−</sup> OT-1 cells by TGFβ in a dose-dependent manner after activation with N4 $10^{-8}$ M TGFβ for 24 h. **d** FoxP1 induction was comparable in the presence of N4 $10^{-8}$ M in control and *Ptpn22*<sup>−/−</sup> OT-1 cells ±5 ng/ml TGFβ for 24 h. **e** Intracellular phosphorylation of Zap70, TCR zeta and ERK in OT-1 cells after TCR stimulation with N4 $10^{-6}$ M ± 5ng/ml TGFβ for 0–10 min. **f** Co-immunoprecipitation of Ptpn22 and Csk in OT-1 cells pretreated ± TGFβ for 2.5 h. Data are from one experiment representative of at least three experiments (mean and s.d. of three replicates). Non significance (ns) as determined using two-way ANOVA with Tukey's post test for multiple comparisons

pathways or FoxP1-dependent processes, as these are unaffected in *Ptpn22*<sup>−/−</sup> T cells, and conversely that TGFβ does not directly affect the function of PTPN22 in regulating T cell signaling.

**IL-2 protects against TGFβ-mediated suppression.** Our data suggested that indirect mechanisms were likely to underlie the ability of *Ptpn22*<sup>−/−</sup> T cells to resist TGFβ inhibition. Serendipitously, we noted that when control and *Ptpn22*<sup>−/−</sup> OT-1 T cells were activated with T4 peptide as a 1:1 mix in the same culture wells, the inhibitory effect of TGFβ on control T cell proliferation was abrogated (Fig. 5a). These data suggested that *Ptpn22*<sup>−/−</sup> T cells secreted or expressed a factor that "rescued" control cells from the effects of TGFβ in a co-culture system. Culture of control and *Ptpn22*<sup>−/−</sup> T cells in transwells also abrogated the inhibitory effect of TGFβ on control T cell proliferation, indicating that a soluble factor was responsible for the protective effect (Fig. 5a, b). TGFβ was shown to limit T cell IL-2 production[20,28,29] and a protective effect of IL-15 and other

common gamma chain cytokines on TGFβ suppression has been reported[7,30]. We found that, compared with control cells, *Ptpn22*<sup>−/−</sup> OT-1 T cells secreted significantly greater quantities of IL-2 in response to both strong N4 and weak T4 agonist peptides (Fig. 5c, ***$p < 0.001$ and ****$p < 0.0001$ using two-way ANOVA with Tukey's post test). Importantly, TGFβ treatment almost completely blocked control but not *Ptpn22*<sup>−/−</sup> OT-1 T cell IL-2 production (Fig. 5c).

Two distinct approaches confirmed the importance of elevated TCR-induced IL-2 production for the ability of *Ptpn22*<sup>−/−</sup> T cells to resist TGFβ-mediated inhibition. First, canonical IL-2-dependent signaling pathways were blocked using the inhibitor tofacitinib, which inhibits Jak3 in addition to other members of the Jak kinase family. Tofacitinib markedly reduced the ability of *Ptpn22*<sup>−/−</sup> OT-1 T cells to withstand TGFβ-mediated inhibition, indicating that IL-2 cytokine receptor signaling was important for the resistance of *Ptpn22*<sup>−/−</sup> cells to TGFβ (Fig. 5d). Second, the addition of recombinant IL-2 to T cell cultures prevented the inhibitory effects of TGFβ on OT-1 T cell proliferation when

added at either d0 or on d2 (Fig. 5e). Recovery of proliferation was dose dependent (Fig. 5f) confirming that insufficiency of IL-2 production in the presence of TGFβ accounted for the inhibition of OT-1 T cell proliferation. Together, these data strongly suggest

that elevated IL-2 production in response to TCR triggering enables $Ptpn22^{-/-}$ T cells to overcome the inhibitory effects of TGFβ.

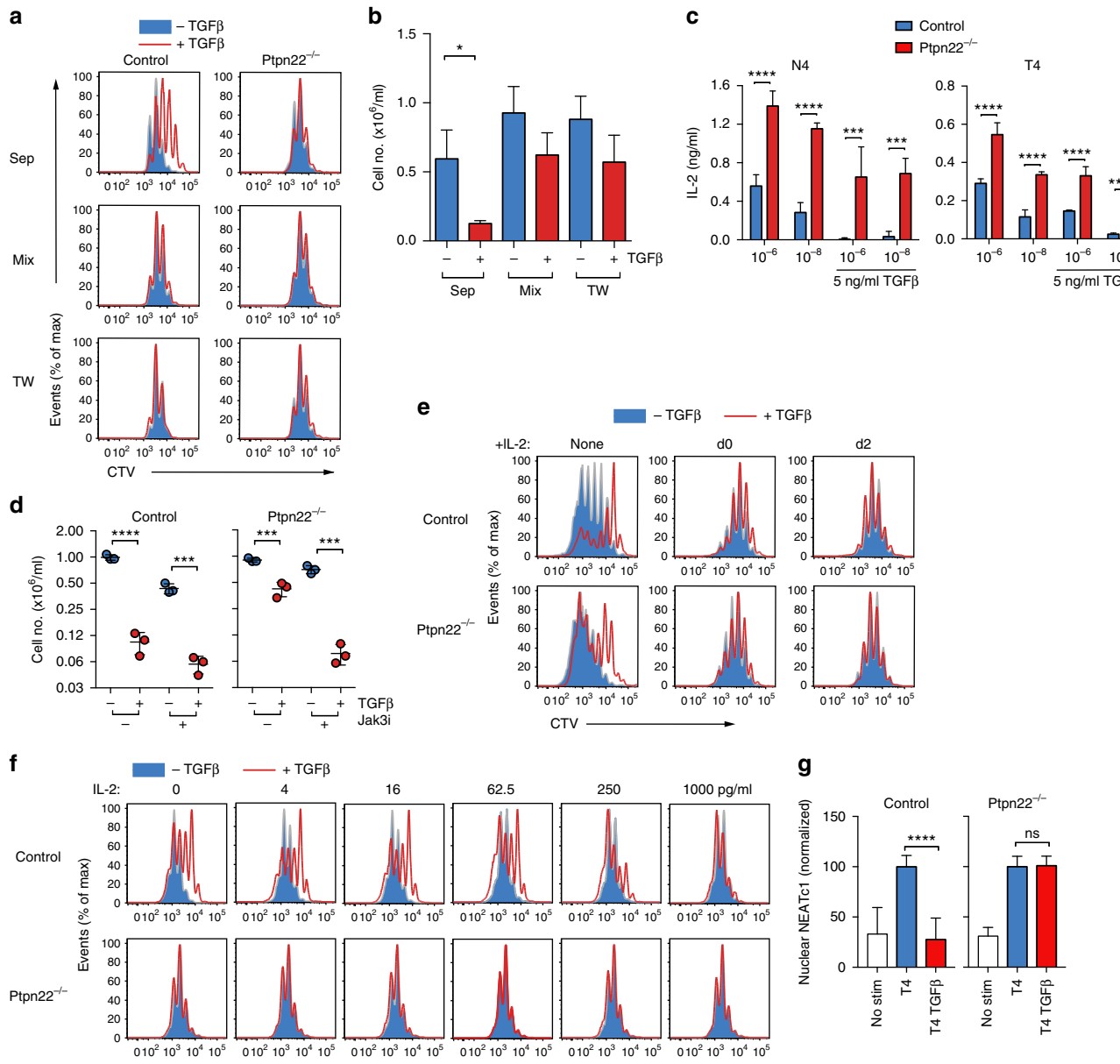

**Fig. 5** Increased IL-2 production from $Ptpn22^{-/-}$ OT-1 cells overcomes TGFβ-mediated suppression. **a** Proliferation of CTV-labeled control and $Ptpn22^{-/-}$ OT-1 cells stimulated with $10^{-8}$ M N4 for 3 days ±TGFβ (5 ng/ml) cultured in separate wells, in the same well (mix) or in transwells. Representative histograms showing CTV dilution are shown. **b** Quantification of recovered control cells. Bars represent mean ± s.e.m. of six individual animals pooled from two separate experiments, *$p < 0.05$ as determined using Students $t$ test. **c** IL-2 was quantified by ELISA in supernatants of control and $Ptpn22^{-/-}$ OT-1 cells stimulated for 24 h with N4 or T4 ($10^{-6}$, $10^{-8}$ M) in the presence of anti-CD25 Ab (10 μg/ml) TGFβ (5 ng/ml). Bars represent triplicate values from groups of three animals, error bar represents s.d. **d** Jak3 signaling is required to overcome TGFβ-mediated suppression. Control and $Ptpn22^{-/-}$ OT-1 cells stimulated with N4 ($10^{-6}$ M) ± TGFβ (5 ng/ml) were cultured for 3d with or without JAK3 inhibitor added for the last 24 h of culture. Live cell recovery is shown, dots represent triplicate values of pooled mice, line represents mean and error bars s.d. **e** Supplementation with IL-2 as late as d2 of culture can override TGFβ-mediated suppression. Control and $Ptpn22^{-/-}$ OT-1 cells stimulated with T4 peptide ($10^{-6}$ M) for 3d, ±TGFβ (5 ng/ml) and ±IL-2 (1 ng/ml) added on d0 or d2. Representative histograms of CTV dilution are shown. **f** Reversal of TGFβ-mediated suppression by titrating IL-2. Control and $Ptpn22^{-/-}$ OT-1 cells stimulated with N4 ($10^{-6}$ M) ±TGFβ (5 ng/ml) were cultured for 3d either without or with the addition of exogenous IL-2 at varying concentrations (0–1000 pg/ml). Representative histograms of CTV dilution are shown. Data **c–f** is representative of at least three experiments. ***$p < 0.001$ and ****$p < 0.0001$, as determined using two-way ANOVA with Tukey's post test for multiple comparisons. **g** $Ptpn22^{-/-}$ OT-1 cells are resistance to TGFβ inhibition of NFATc1 translocation. Control and $Ptpn22^{-/-}$ OT-1 cells stimulated with T4 ($10^{-6}$ M) ±TGFβ for 4 h were lysed and nuclear extracts analysed for NFATc1 expression. Data are pooled and normalized from three separate experiments. Error bars represent s.e.m. and ****$p < 0.0001$, as determined using Student's $t$ test

Previous studies have suggested that TGFβ interferes with TCR-induced $Ca^{2+}$ signaling and subsequent NFAT activation, thereby limiting T cell activation and IL-2 production[31]. We have shown previously that $Ca^{2+}$ fluxes in naive T cells were not influenced by the absence of Ptpn22[17]. However, in vitro measurement of $Ca^{2+}$ flux in primary naive T cells by FACS requires very acute and strong stimulation, e.g., in the OT-1 model by using multivalent peptide:MHC complexes containing strong agonist peptide N4[32]. Under these stimulation conditions, we saw no reduction of $Ca^{2+}$ signals in control OT-1 T cells in the presence of TGFβ (Supplementary Fig. 1). However, we reasoned that looking more downstream, e.g., at NFAT nuclear translocation might reveal some differences. Indeed, we observed significant inhibition of NFAT nuclear translocation in the presence of TGFβ when control OT-1 T cells were stimulated with T4 peptide (Fig. 5g, ****$p < 0.0001$ using Student's $t$ test). In contrast, T4-induced NFATc1 nuclear translocation in stimulated $Ptpn22^{-/-}$ CD8 T cells was remarkably resistant to suppression by TGFβ (Fig. 5g). These data corroborate the finding that TGFβ suppresses control CD8 T cell proliferation by interfering with IL-2 production through inhibition of NFAT activation, and indicate that in the absence of PTPN22, this inhibitory pathway is circumvented.

## Discussion

The ability of T cells to integrate and respond appropriately to a complex network of signals triggered by antigen receptor engagement, cytokines, chemokines, and other environmental signals is critical to the outcome of immune responses. Furthermore, in order to maintain tolerance, cell-intrinsic inhibitory mechanisms and the action of suppressive cytokines, including TGFβ prevent T cell activation by low-affinity self-antigens. In the current work, we found that altering the sensitivity of TCR signaling, by deletion of the autoimmune-associated phosphatase PTPN22, further impacted upon the sensitivity of CD8[+] T cells to the suppressive effects of TGFβ. Our results describing the consequent imbalance in positive and negative signaling have important implications for our understanding of the mechanisms regulating autoimmune T cells and may also have practical applications in tumor therapy. In this regard, we showed that the combination of enhanced TCR sensitivity and subsequent resistance to TGFβ inhibition enabled $Ptpn22^{-/-}$ T cells to have potent anti-tumor effector function in vivo.

A key mechanism underlying resistance to TGFβ suppression was the ability of $Ptpn22^{-/-}$ T cells to produce IL-2 more abundantly than control T cells. Other studies have shown that $Ptpn22^{-/-}$ CD4[+] T cells make more cytokines, including IL-2, than WT T cells[33,34] and we show that is true for $Ptpn22^{-/-}$ CD8[+] cells also. Previous data indicated that Smad3-dependent inhibition of Il2 transcription and protein expression was key to TGFβ-induced suppression of TCR-induced T cell proliferation[35]. Consistent with this, in our studies, suppression of peptide-induced control OT-1 T cell proliferation could be reversed by addition of recombinant IL-2 to cultures. It is worthy of note that PTPN22 does not directly regulate IL-2[33] or TGFβ receptor signaling pathways. Therefore, it is likely that the ability of $Ptpn22^{-/-}$ T cells to withstand TGFβ-induced suppression is a result of quantitative effects of elevated IL-2 production rather than qualitative effects on IL-2R or TGFβR signaling. In addition, our data highlight the complexity of the interplay between IL-2 and TGFβ signaling on T cell responses. In this regard, IL-2 abrogates the pro-inflammatory effects of TGFβ on CD4[+] Th17 differentiation and instead favors inducible Treg expansion[36]. By contrast, our data show that elevated IL-2 production by CD8[+] T cells abrogates suppressive effects of TGFβ and instead favors a pro-

inflammatory response. Thus, the effects of these two cytokines in vivo are likely to be highly context dependent.

The mechanisms by which TGFβ inhibits TCR signaling and T cell function are not fully understood. An early report suggested that, in CD4[+] T cells, TGFβ signals interfere with very early TCR-induced Tek and ERK kinase activation and NFAT nuclear translocation[31]. Our data support this view as we saw clear inhibition of NFAT translocation in control cells incubated with TGFβ, whereas $Ptpn22^{-/-}$ T cells were resistant to this inhibiiton. Analysis of more upstream pathways did not reveal more specifically how TGFβ inhibition intersects TCR signaling, as we saw no change in TCRζ, Zap70, or MAPK signaling in WT T cells stimulated in the presence or absence of TGFβ. However, these assays utilize strong stimulation of the TCR over very short time frames and may therefore lack the sensitivity to observe TGFβ inhibition. More recently, TGFβ-mediated T cell suppression has been linked to effects on the mTOR pathway[37]. In those studies, suppressive effects of TGFβ on mTOR signaling were assessed following 36 h (or more) of T cell stimulation, time points at which autocrine IL-2 signals contribute to mTOR activation[38]. Therefore, it is possible that TGFβ inhibition of mTOR signals at these later time points might, at least in part, be explained by reduced IL-2 secretion. In NK cells, TGFβ also suppresses mTOR activation with potent downstream effects on cellular metabolism[39]. Given the importance of shifts in cellular metabolism in the generation and differentiation of effector T cells, it will be interesting to determine whether TGFβ also impedes these processes in T cells. A recent study indicated that FoxP1 expression was required for TGFβ-mediated inhibition of anti-tumor CD8[+] T cells[25]. However, our data showed that $Ptpn22^{-/-}$ and control OT-1 T cells expressed equivalent levels of FoxP1. Therefore, it is likely that FoxP1 expression is necessary but not sufficient to mediate TGFβ suppression.

Our experiments provide new insight into the mechanisms by which PTPN22 dampens T cell activation. Loss of PTPN22 function increases the chance of T cells responding to weak, self-antigens and this enhanced TCR sensitivity is compounded by a knock-on effect on the inhibitory TGFβ pathway. While these effects are likely to be detrimental in the context of autoimmune disease, they also raise the possibility that GWAS-identified, autoimmune susceptibility loci can be manipulated to improve the intrinsic efficacy of adoptive cell therapy (ACT) for cancer. In this regard, we showed that $Ptpn22^{-/-}$ T cells were superior to control cells in responding to both high- and low-affinity TSA and in mediating tumor rejection in vivo, suggesting that targeting PTPN22 may be a viable approach in T cell immunotherapies. We reason that in future studies, by restricting loss of PTPN22 to tumor-specific T cells, undesirable autoimmunity will be minimized. Despite the recent success of chimeric antigen receptors[40], a major advantage of retaining TCR expression for ACT is that many TSAs are intracellular proteins that are only presented in the context of MHC. However, TSAs are frequently too weak to stimulate effective T cell responses, but by manipulating the intracellular signaling machinery, we show that it is possible to harness traits that are undesirable in autoreactive T cells, such as reactivity to weak antigens and resistance to suppressive cytokines, for the benefit of anti-tumor responses.

## Methods

**Mice.** Rag-1[−/−] OT-I CD45.1[+/+] [40], $Ptpn22^{-/-}$ Rag-1[−/−] OT-I CD45.2[+/+] [18] and CD45.1[+/−]CD45.2[+/−] mouse strains were bred and maintained under specific and opportunistic pathogen-free conditions at the Universities of Edinburgh and Leeds. Age-matched (6–12 weeks) and sex-matched mice were used in all experiments. Animal procedures were approved under a UK Home Office project license and were performed in compliance with the ethical guidelines of the Universities of Edinburgh and Leeds.

**Flow cytometry and antibodies**. The following conjugated antibodies were used: phycoerythrin–indotricarbocyanine–anti-CD8β (clone eBioH35-17.2, cat 15266777, 1:400 dilution), phycoerythrin–anti-CD25 (clone PC61.5, cat 15298489, 1:200 dilution), allophycocyanin–eFluor 780–anti-CD44 (clone IM7, cat 15351620, 1:400 dilution), fluorescein isothiocyanate–anti-TCRαβ (clone H57-597, cat 11-5961-81, 1:200 dilution), phycoerythrin–anti-T-bet (clone 4B10, cat 12-5825-80, 1:100 dilution), eFluor 450–anti-IRF4 (clone 3E4, cat 15519846, 1:200 dilution), eFluor 450–anti–granzyme B (clone NG2B, cat 15371830, 1:100 dilution), (all from eBioscience); Alexa Fluor 488–anti-CD25 (clone PC61.5, cat 102018, 1:200 dilution), allophycocyanin–anti-CD45.1 (clone A20, cat 110713, dilution 1:200), fluorescein isothiocyanate–anti-CD45.1 (clone A20, cat 110718, dilution 1:200), phycoerythrin–anti-CD45.2 (clone 104, cat 109807, dilution 1:200), Brilliant Violet 421–anti-CD45.2 (clone 104, cat 109832, dilution 1:200), Alexa Fluor 488–anti-IFN-γ (clone XMG1.2, cat 505813, dilution 1:200) (all from BioLegend). Unconjugated rabbit antibodies to c-Myc (clone D84C12, cat 5605, dilution 1:50), phospho-Smad2 (Ser465/467)/Smad3 (Ser423/425) (clone D27F4, cat D27F4, dilution 1:100), FoxP1 (clone D35D10 XP, cat 4402, dilution 1:200), phospho-Zap70 (Tyr493)/Syk (Tyr526) Y493 (cat 2704, dilution 1:100), phospho-p44/42 MAPK (Erk1/2) (Thr202/Tyr204) (clone D13.14.4E, cat 4370, dilution 1:100) were from Cell Signaling Technologies and anti-CD3 zeta (phospho Y83) (clone EP776 (2)Y, cat ab68236, dilution 1:50) was from Abcam; all were counterstained with goat anti-rabbit (A21244; Molecular Probes). Phycoerythrin anti-mouse TGFbeta RII (cat FAB532P, 1:50) and isotype control were from R&D systems. Live/Dead Aqua and Cell Tracer Violet (CTV) dyes were from Life Technologies. For intracellular staining, cells were fixed in Phosflow Lyse/Fix Buffer 1 (BD) or FoxP3 Fix/Permeabilization Buffer (eBioscience) before staining with antibody in Permeabilization/Wash buffers (eBioscience). For phospho-SMAD2/3, staining cells were fixed with 2% paraformaldehyde and permeabilized with methanol prior to staining. Samples were acquired with a MACSQuant flow cytometer (Miltenyi) and data were analyzed with FlowJo software (Treestar). Division index is the average number of cell divisions that a cell in the original population has undergone (i.e., includes the undivided peak).

**T cell culture and stimulation**. T cells from the lymph nodes of wild-type and *Ptpn22*$^{-/-}$ mice were cultured in Iscove's Modified Dulbecco's Medium (IMDM, Invitrogen) supplemented with 10% FCS, L-glutamine, antibiotics and 50 μM 2-mercaptoethanol. The agonist peptide SIINFEKL (N4) and partial agonist SIITFEKL (T4) (Peptide Synthetics) were added to culture medium at concentrations stated in figure legends. Recombinant human TGFβ (R&D) was added to cultures at 5 ng/ml unless otherwise stated and, where indicated, a TGFβ inhibitor (SB431542, 2.5 μM, Abcam), Jak3 inhibitor (tofacitinib, 200 nM, Tocris Bioscience), or recombinant human IL-2 (Peprotech) were also added. Transwell experiments were performed using HTS Transwell 96 system 0.4 μm pore polycarbonate membrane (Corning). For measurement of IL-2, cells were stimulated with peptide ± TGFβ in the presence of blocking CD25 mAb (Biolegend) in order to prevent IL-2 consumption. IL-2 in culture supernatants was measured using the Mouse IL-2 ELISA Ready-SET-Go!® kit (eBioscience).

**Immunoprecipitation and western blot**. T cells were generated in vitro by stimulating OT-1 control or *Ptpn22*$^{-/-}$ cells for 2d with 0.01 μM N4, washed, then expanded for a further 6d in 20 ng/ml recombinant human IL-2 (Peprotech) followed by treatment for 2.5 h ± TGFβ (5 ng/ml). Cells were lysed in 1% TritonX-100, 0.5% n-dodecyl-b-D-maltoside, 50 mM Tris-HCl, 150 mM NaCl, 20 mM EDTA, 1 mM NaF, and 1 mM sodium orthovanadate containing protease inhibitors. Samples were immunoprecipitated with anti-Ptpn22 antibody (clone D6D1H, Rabbit mAb Cell Signaling Technology) coupled to Dynabeads prot G (10003D; Life Technology). Western blots were performed on samples and membranes were probed with anti-Ptpn22 rabbit mAb (clone D6D1H, cat CS14693S, dilution 1:1000 from Cell Signaling Technology) and anti-murine CSK antibody (clone 52, cat 610079, dilution 1:1000, from BD Transduction Labs) and proteins detected with secondary Abs and visualized using infrared imaging system (Odyssey; LI-COR Biosciences). For uncropped western blot image, see Supplementary Fig. 2.

**NFAT translocation assay**. Control and *Ptpn22*$^{-/-}$ OT-1 cells ($2 \times 10^6$ cells) were stimulated with T4 peptide (T4$^{-6}$ M) ± TGFβ for 4 h at 37 °C. Cells were then lysed and nuclear extracts prepared and analysed for NFATc1 using the TransAM NFATc1 Transcription Factor assay kit according to the manufacturer's instructions (Activemotif).

**Calcium flux**. OT-1 cells ($1 \times 10^7$ cells) were prestained with anti-CD8β (clone eBioH35-17.2, cat 15266777, 1:400 dilution) in order to identify cell populations prior to loading with 2 μM Indo-1 AM (Life Technologies) for 40 min at 37 °C in PBS. Cells were resuspended in pre-warmed IMDM containing 1% FCS and supplemented with 2.5 mM CaCl$_2$ and 2.8 mM MgCl$_2$. The baseline Ca$^{2+}$ levels were measured for 1 min before PE-labeled N4 dextramer (Immudex, Denmark) was added to stimulate the cells. Some samples were pretreated with TGFβ (5 ng/ml) 5 min before the addition of dextramer. Ionomycin (1 μg/ml) was added as a positive control for measuring saturated Ca$^{2+}$ levels. Data are represented as the

ratio of 398 nm (Indo-1 bound to Ca$^{2+}$)/482 nm (unbound Ca$^{2+}$) in OT-1 cells by flow cytometry and all data were acquired on an LSRII (BD Bioscience).

**Cell lines**. ID8 ovarian carcinoma cells were a gift from Dr K. Roby, University of Kansas, USA[23]. EL4-OVA cells were provided by Dr K. Okkenhaug, Babraham Institute, UK and EL4-OVAsTGFβRII cells were provided by Prof. J. Massagué, Memorial Sloan Kettering Cancer Institute, New York, USA. All cell lines were negative for mouse pathogens, including mycoplasma contamination (IMPACT IV, IDEXX BioResearch).

**ID8 tumor experiments**. ID8 ovarian carcinoma cells[23] were transduced with retroviral constructs to express the ova-variant protein T4 (aminoacids 197–387). ID8-T4 cells were maintained in IMDM supplemented with 10% FCS, 50 μM 2-mercaptoethanol, L-glutamine and antibiotics (100 U penicillin, 100 mg/ml streptomycin, 0.5% ciproxin). For co-culture experiments, sub-confluent ID8-T4 cells were harvested and transferred to 48-well tissue culture plates at $2 \times 10^4$ cells/well. Cells were allowed to settle and adhere to culture plates for 4 h then OT-I T cells were added ($2 \times 10^5$/well) and cells co-cultured for 48 h. For analysis of T cell intracellular cytokine expression, Brefeldin A (2.5 mg/ml, Sigma) was added to wells for the last 6 h of culture prior to staining using the antibodies described above.

For in vivo T cell expansion studies, abdominal tumors were established using $10^7$ ID8 cells injected i.p. to recipient CD45.1$^{+/-}$ CD45.2$^{+/-}$ mice. After 28 days, OT-I CD45.1$^{+/+}$ and *Ptpn22*$^{-/-}$ OT-I CD45.2$^{+/+}$ T cells were mixed 1:1, stained with CTV dye (Molecular Probes) and transferred i.p. to tumor-bearing or control mice. After 3 days, peritoneal lavages were performed and tumor-induced T cell proliferation assessed as described above.

For tumor rejection studies, ID8-T4 cells were stably transduced with Firefly luciferase-expressing lentiviral vector (kindly provided by M. Lorger, University of Leeds) (ID8-T4-fluc2). Abdominal tumors were established by injection of $5 \times 10^6$ ID8-T4-fluc2 cells i.p. into B6 recipients. Quantification of tumor growth was performed via non-invasive bioluminescence imaging using IVIS Spectrum and Living Image software (Perkin Elmer) on day 11 prior to adoptive T cell transfer. T cells for transfer were generated in vitro by stimulating OT-1 control or Ptpn22$^{-/-}$ cells for 2 days with 0.01 μM N4, washed, then their populations expanded for a further 4 days in 20 ng/ml recombinant human IL-2 (Peprotech). On day 12, $10^7$ OT-1 control or OT-1 Ptpn22$^{-/-}$ effector CTLs cells were injected i.p. into recipient tumor-bearing mice and tumor growth monitored by bioluminescence imaging at various time points thereafter.

**EL4 tumor experiments**. EL4-OVA and EL4-OVAsTGFβRII cells were maintained in IMDM supplemented with 10% FCS, 50 μM 2-mercaptoethanol, L-glutamine and antibiotics (100 U/ml penicillin, 100 μg/ml streptomycin, 400 μg/ml geneticin (G418) and 0.5% ciproxin). Tumor cells in log-phase growth were resuspended in PBS and $10^6$ cells were injected subcutaneously into the right dorsal flank. Expression of SIINFEKL/H-2K$^b$ pMHC complexes in >97% EL4 cells was verified by flow cytometry using anti-SIINFEKL/H-2K$^b$ antibody (25 D1.6). Tumor growth was monitored at least twice weekly using calipers with tumor volume calculated as length × width × the average of length and width measurement to approximate the total volume of the subcutaneous tumor mass (mm$^3$)[41]. Five days after tumor cell injection, a time point at which 30–50% of animals presented with palpable tumor mass, recipient mice were randomly divided into three groups. Control mice received no adoptive cell transfer, whereas other groups received adoptive transfer of $10^5$ control or *Ptpn22*$^{-/-}$ OT-1 T cells by intravenous injection. Following 7–10 days, mice were killed, tumors resected and tumor mass determined.

**Statistical analysis**. Prism software was used for Student's *t* test (paired or unpaired; two tails), two-way ANOVA with Tukey's post test for multiple comparisons and $\chi^2$-test with Fisher's exact post test.

**Data availability**. The data that support the findings of this study are available within the article and its Supplementary Information files and from the corresponding authors upon reasonable request.

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

## Acknowledgements

We thank J. Massagué for providing EL4-OVA STβRII cells and K. Okkenhaug for providing EL4-OVA cells; M. Lorger and T. Andreou for providing Firefly luciferase-expressing lentiviral vector, Judi Allen and Paul Travers for critically reading the manuscript; David Wright and Xiaoyan Zou for technical assistance; Graeme Cowan and Margo Chase-Topping for statistical help. The work was funded by Wellcome Trust Investigator Award, WT096669AIA, to R.Z., a strategic award, WT095831, for the Centre for Immunity, Infection and Evolution, and a Cancer Research UK grant (23269) to R.S.

## Author contributions

R.J.B., R.J.S., and R.Z. designed the study; R.J.B., C.G., M.R., and R.J.S. carried out the experiments; D.Z. developed and supplied key reagents; R.J.B., R.J.S., and R.Z. wrote the manuscript; R.J.S. and R.Z. contributed key concepts and share equal senior authorship.

## Additional information

**Competing financial interests:** The authors declare no competing financial interests.

