## [Peer Review File · Nature Communications]

Reviewer #1:

This manuscript provides novel information suggesting that knocking down PTPN22 in T cells used for adoptive cell transfer (ACT) therapy would result in increased antitumor activity mediated by insensitivity to TGFbeta. This is a clinically translatable approach.

Major comments:

The data is based on OT-1 ACT therapy using two tumor targets expressing OVA. It would be desirable to test the generality of this approach using a tumor that expresses an endogenous tumor antigen as opposed to a strong xenoantigen. This could be the pmel-1/B16 model or a similar model where the tumor cells are unmodified.

How often are cancers positive or negative for TGFbeta? The data seems to imply that only cancers that express significant TGFbeta in a cancer cell-intrinsic fashion may respond to this approach. The authors could analyze publicly available transcriptome data from the cancer cell encyclopedia (CCLE) or the cancer genome atlas (TCGA) databases.

Minor comments:

Ideally, as the only clinically relevant way to test this concept would be using ex vivo modification of T cells for ACT, additional testing in a model of T cell receptor (TCR) or chimeric antigen receptor (CAR) engineering of T cells with concomitant PTPN22 silencing by shRNA or Crispr/Cas9 would significantly strengthen the article.

To conclusively demonstrate that the PTPN22 null T cells expand better in vivo it would have been desirable to genetically label them with firefly luciferase and use a CCD camera for bioluminescence imaging.

Reviewer #2:Expert in Cancer immune microenvironment
(Remarks to the Author):

In the present manuscript, Brownlie et al showed that the tyrosine phosphatase Ptpn22 has a relevant role in decreasing T cell proliferation in response to both weak and strong affinity antigens. Ptpn22 deficient T cells produced higher levels of IL-2 and granzyme B, and showed higher proliferation rates after TGF-b stimulation in vitro or in an in vitro model of TGF-b-secreting tumor.

The experiments are well developed and the manuscript reads well, but some additional points should be addressed. The contribution of Ptpn22 in controlling IL-2 production by T cells has been previously shown by Maine et al. 2016, and similarities in between the two works should be discussed.

Although it is clear that IL-2 production by Ptpn22^{-/-} T cells favors cell proliferation in a context of TGF-b-induced suppression, the mechanisms involved the regulation of IL-2 are not very clear. It is critical to show whether TGF-b directly regulates Ptpn22 activation. If this is not true, would Ptpn22^{-/-} T cells be refractory to other suppression pathways?

In the figure 3d, the authors showed increased granzyme B production by Ptpn22^{-/-} T cells cocultured with ID8 tumor cells. These findings raise the question whether Ptpn22 also controls activation status of the T cells. In figure 3 the authors also showed that TGF-b inhibitor potentiates granzyme B production and improves control of the tumor growth. Thus, analyzing the proportion

of regulatory T cells in wild type and Ptpn22^{-/-} T cells could shed light the mechanisms regulating such responses.

We thank the reviewers for their helpful comments and herewith submit a revised manuscript (changes highlighted in yellow) and accompanying figures which we trust will address their concerns. We have also added two more authors as they undertook some of the additional experiments required for the resubmission. We hope you will find the revised manuscript suitable for publication in Nat Communications.

Detailed answers to the reviewers' comments are given in red below.

Reviewer #1:

(Remarks to the Author):

This manuscript provides novel information suggesting that knocking down PTPN22 in T cells used for adoptive cell transfer (ACT) therapy would result in increased antitumor activity mediated by insensitivity to TGFbeta. This is a clinically translatable approach.

Major comments:

The data is based on OT-1 ACT therapy using two tumor targets expressing OVA. It would be desirable to test the generality of this approach using a tumor that expresses an endogenous tumor antigen as opposed to a strong xenoantigen. This could be the pmel-1/B16 model or a similar model where the tumor cells are unmodified.

We appreciate that the reviewer has concerns that using the N4 peptide as a TSA introduces a strong xenoantigen in the EL4 tumor model – although virally induced (eg EBV & HPV) tumors also express strong xenoantigens, so arguably the model is valid. We considered it critical to compare T cells with an identical TCR ± PTPN22 in this study, as otherwise T cells expressing different affinity TCRs could have been activated in control and KO mice in response to tumor, and this might have had a profound influence on how readily the cells were inhibited by TGFβ. By using OT-1 cells we can ensure the affinity of the TCR is constant and does not contribute to whether or not the cells are suppressed by TGFβ. In order to backcross our mutation onto the TCR transgenic mouse with specificity for pmel would require acquisition of these lines followed by at least

one year of backcrossing to get the appropriate genotypes, so as an alternative to address the reviewer's concern we have generated a firefly luciferase-expressing variant of the ID8 tumor cell line expressing the T4 epitope which has a functional avidity for OT-1 T cells that is at least 70-fold less than N4. It was shown (Daniels, et al *Nature*, 444(7120), 724–729) that T4 can positively select OT-1 cells in the thymus, so it is of comparable affinity to self-peptides and thus better reflects a weak tumor antigen. Using this new tool, we now show that ID8-T4 tumors are also controlled more effectively by adoptive cell transfer of *Ptpn22*^{-/-} T cells as compared to control T cells (Fig 3e).

How often are cancers positive or negative for TGFbeta? The data seems to imply that only cancers that express significant TGFbeta in a cancer cell-intrinsic fashion may respond to this approach. The authors could analyze publicly available transcriptome data from the cancer cell encyclopedia (CCLE) or the cancer genome atlas (TCGA) databases.

This is an interesting point and CCLE confirms the majority of tumors express TGFβ. Furthermore, in addition to tumor cell-intrinsic TGFβ expression, a range of tumor-infiltrating cells such as MDSCs may also contribute to high levels of TGFβ with the tumor microenvironment. We have clarified these points with additional text in the introduction (p4, revised manuscript). In addition to their resistance to TGFβ suppression, we have also shown that *Ptpn22*^{-/-} CD8 T cells respond better to weak antigens (Salmond, R. J., et al 2014. *Nature Immunology*, 15(9), 875–883) such as T4 expressed by ID8 tumors (Fig 3c) and are resistant to suppression by WT Tregs (Brownlie et al. 2012 *Sci Sig* 5:ra87) which may also contribute to their improved anti-tumor efficacy.

Minor comments:

Ideally, as the only clinically relevant way to test this concept would be using ex vivo modification of T cells for ACT, additional testing in a model of T cell receptor (TCR) or chimeric antigen receptor (CAR) engineering of T cells with concomitant PTPN22 silencing by shRNA or Crispr/Cas9 would significantly strengthen the article.

We are indeed undertaking these kinds of analyses with human T cells transduced with anti-tumor TCRs and Crispr, but that is an entirely new study and will take several years to complete so is outwith the scope of this manuscript.

To conclusively demonstrate that the PTPN22 null T cells expand better in vivo it would have been desirable to genetically label them with firefly luciferase and use a CCD camera for bioluminescence imaging.

We respectfully disagree, as dilution of CTV dye as shown in Fig 3d gives a very accurate division rate for the population by allowing us to track proliferation of individual cells by FACS, which is not achievable by luciferase imaging. In addition, and in combination with the use of allelic markers, we can track the two populations in an identical environment in individual mice.

Reviewer #2

In the present manuscript, Brownlie et al showed that the tyrosine phosphatase Ptpn22 has a relevant role in decreasing T cell proliferation in response to both weak and strong affinity antigens. Ptpn22 deficient T cells produced higher levels of IL-2 and granzyme B, and showed higher proliferation rates after TGF- β stimulation in vitro or in an in vitro model of TGF- β -secreting tumor.

The experiments are well developed and the manuscript reads well, but some additional points should be addressed. The contribution of Ptpn22 in controlling IL-2 production by T cells has been previously shown by Maine et al. 2016, and similarities in between the two works should be discussed.

The reviewer is correct and we have now acknowledged this in the discussion, although the Maine et al, and indeed the original Hasegawa et al study, both showed more IL-2 was produced by PTPN22-KO CD4 T cells, rather than CD8 T cells, as we show here.

Although it is clear that IL-2 production by Ptpn22^{-/-} T cells favors cell proliferation in a context of TGF- β - induced suppression, the mechanisms involved the regulation of IL-2 are not very clear. It is critical to show whether TGF- β directly regulates Ptpn22 activation. If this is not true, would Ptpn22^{-/-} T cells be refractory to other suppression pathways?

That TGF β influences IL-2 production by T cells was first reported in the early 1990s and since then there has been rather little published that enlightens the mechanism of how this works despite its obvious importance. We do not think TGF β impacts Ptpn22 function directly, as we can see no changes in early signaling in the presence of TGF β , which is where Ptpn22 seems to have the most influence (new Fig 4e) nor in the association of Ptpn22 with Csk (new Fig 4f). However TGF β likely reduces TCR signals to some degree as nuclear translocation of NFAT is inhibited in the presence of TGF β in WT cells (Fig 4g). That might suggest TGF β impacts upstream pathways such as Ca signaling, however, this effect is likely to be quite subtle. In contrast the assays required to visualize these kinds of early signals, such as Ca flux, need heavy receptor cross-linking to work, and under these conditions we see no change in the

presence of TGF β . For this reason we went downstream to look at NFAT, as the magnitude of small upstream changes can be amplified at the level of transcription factor activation.

We think it is likely that the Ptpn22^{-/-} T cells are more refractory to other suppressive pathways as we have shown that they resist suppression by WT Treg better than WT CD8 T cells, although Ptpn22^{-/-} Tregs will suppress Ptpn22^{-/-} T cells effectively (Brownlie et al. 2012 Sci Sig 5:ra87).

In the figure 3d, the authors showed increased granzyme B production by Ptpn22^{-/-} T cells cocultured with ID8 tumor cells. These findings raise the question whether Ptpn22 also controls activation status of the T cells. In figure 3 the authors also showed that TGF-b inhibitor potentiates granzyme B production and improves control of the tumor growth. Thus, analyzing the proportion of regulatory T cells in wild type and Ptpn22^{-/-} T cells could shed light the mechanisms regulating such responses.

The experiments in Fig 3 were done with CD8⁺ TCR transgenic LN cells on a Rag-KO background so these contain ~97% CD8 T cells and a few DC, but no CD4 T cells or Treg are involved in these assays, apologies if that was not clear. However the reviewer is correct in that we have previously shown there are more Tregs in KO mice which was why we always transferred T cells for the tumor studies, so that the only difference between the groups was the genotype of the adoptively transferred CD8 cells.

Reviewer #1 (Remarks to the Author):

The authors have not addressed the main concerns of the generability of the findings beyond the OT-1 ACT model. It would have been desirable to expand to other models, which could be done by generating knock out bone marrow chimeras as opposed to having to generate and cross new genetically modified mice. Also, analysis of publically available gene expression datasets to study the frequency of TGF β , PTPN22 and immune responses in human cancers would have further strengthened the article.

Reviewer #2 (Remarks to the Author):

Authors wrote "However TGF β likely reduces TCR signals to some degree as nuclear translocation of NFAT is inhibited in the presence of TGF β in WT cells (Fig 4g)." I do not see the figure 4g in the manuscript. But other than that, the authors have sufficiently revised the manuscript to answer all this reviewers previous concerns.

In response to the reviewers comments:

Reviewer 1

“The authors have not addressed the main concerns of the generability of the findings beyond the OT-1 ACT model. It would have been desirable to expand to other models, which could be done by generating knock out bone marrow chimeras as opposed to having to generate and cross new genetically modified mice. Also, analysis of publically available gene expression datasets to study the frequency of TGF β , PTPN22 and immune responses in human cancers would have further strengthened the article.”

Unfortunately we cannot combine the PTPN22 knockout mutation with a TCR specific for tumour antigen except by than breeding the two together – making a chimera does not help in this instance. We did explore the publically available gene expression databases for the frequency of cancer cells expressing TGF β and, as we report in the manuscript, find it is very high. We did not look for PTPN22 and immune responses in human cancer, as our study is focused exclusively on isolated CD8 T cells, which might be useful for adoptive cell therapy. PTPN22 is expressed in multiple haematopoietic lineages and the disease-associated allele may, in consequence, have both positive and negative influences on tumour progression (eg if inhibitory monocytes were more active) which would be difficult to disentangle.

Reviewer 2

Authors wrote "However TGF β likely reduces TCR signals to some degree as nuclear translocation of NFAT is inhibited in the presence of TGF β in WT cells (Fig 4g)." I do not see the figure 4g in the manuscript.

Apologies this should have been Figure 5g and this has been corrected.